# Crystallization Behavior of Poly(ε-Caprolactone)-Hollow Glass Microspheres Composites for Rotational Molding Technology

**DOI:** 10.3390/polym14204326

**Published:** 2022-10-14

**Authors:** Adriano Vignali, Roberto Utzeri, Maurizio Canetti, Fabio Bertini

**Affiliations:** 1Istituto di Scienze e Tecnologie Chimiche “Giulio Natta” (SCITEC)-CNR, Via A. Corti 12, 20133 Milano, Italy; 2Istituto di Scienze e Tecnologie Chimiche “Giulio Natta” (SCITEC)-CNR, Via De Marini 6, 16149 Genova, Italy

**Keywords:** polymer composites, polycaprolactone (PCL), glass microspheres, crystallization behavior, rotational molding

## Abstract

Composites suitable for rotational molding technology based on poly(ε-caprolactone) (PCL) and filled with hollow glass microspheres (HGM) or functionalized hollow glass microspheres (HGMf) were prepared via melt-compounding. The functionalization of glass microspheres was carried out by a silanization treatment in order to improve the compatibility between the inorganic particles and the polymer matrix and achieve a good dispersion of glass microspheres in the matrix and an enhanced filler–polymer adhesion. The crystallization behavior of materials was studied by DSC under isothermal and non-isothermal conditions and the nucleating effect of the glass microspheres was proven. In particular, the presence of silanized glass microspheres promoted faster crystallization rates and higher nucleation activity, which are enhanced by 75% and 50%, respectively, comparing neat PCL and the composite filled with 20 wt% HGMf. The crystalline and supermolecular structure of PCL and composites crystallized from the melt was evaluated by WAXD and SAXS, highlighting differences in terms of crystallinity index and structural parameters as a function of the adopted crystallization conditions.

## 1. Introduction

Rotational molding (RM), also known as rotomolding, is a high temperature, low pressure, and low shear process for production of hollow plastic items [1,2,3]. Nowadays, RM is a growing sector of the polymer processing industry. This growth rate is due to the fact that rotational molding offers the opportunity to produce one piece of hollow articles of any dimension that are essentially stress free. The rotomolding process is different from the principal processing techniques for polymers (i.e., injection molding and extrusion). In details, RM can be divided into the following steps. Initially, polymer powder is put in an open mold at room temperature. Afterwards, the mold is closed and placed into an oven where it is heated while being rotated around two normal axes. The speed of rotation is slow, about 10 rpm, and the speeds of the two axes are generally different in order to guarantee a uniform thickness of the molding. Once all of the polymer material has been melted and coated in the inner part of the mold, the rotation goes on as the mold is cooled by air or water spray to promote the polymer solidification. Finally, the molded part is removed. An important peculiarity of RM is that the used cooling rates can cover a wide range (from 5 to 30 °C/min), thus several morphologies can be obtained. Moreover, only the external surface of the rotomolded part is in contact with the mold. Instead, the inner free surface is cooled at a different rate. For these reasons, the microstructure of rotomolded items is peculiar and the solidification stage plays an important role. The morphology strongly influences the product’s performance in terms of the mechanical properties, dimensional stability, barrier properties and chemical resistance according to the application of the rotomolded product.

Most of the thermoplastic materials used in rotomolding application are semicrystalline polymers; in particular, different grades of polyethylene currently represent over 85% of all polymers rotomolded in worldwide production. Thus, crystallization, occurring during the cooling step, is of great importance for the morphology and the resulting properties of the final product. Several studies were focused on polyethylene-based materials in order to investigate crystallization during the RM process [4,5,6,7] and the final properties of the rotomolded composites [8,9,10,11,12,13]. However, the recent growing interest for RM aimed to investigate the crystallization behavior of other polymers such as polypropylene [14,15], polyamide [16], and polylactide [17,18] for application in this technology process.

In a previous paper, we reported on thermal, rheological and mechanical properties of lightweight poly(ε-caprolactone) (PCL) composites with surface modified hollow glass microspheres for use in RM [19]. PCL is a linear aliphatic polyester that has received much attention for biomedical purposes because of its possible application as biocompatible and biodegradable material. In particular, PCL was extensively used in drug delivery systems, medical devices, and tissue engineering scaffolds [20,21]. Moreover, PCL is a semicrystalline polymer whose final mechanical properties depend on crystallinity and morphology. Therefore, the detailed knowledge of the crystallization process is fundamental to develop PCL-based materials with tailored crystalline structure and properties.

Several studies on PCL crystallization kinetics have been reported in the literature. Schick et al. carried out various research on PCL crystallization by differential fast scanning calorimetry demonstrating the potentialities of this calorimetric technique. In particular the authors studied the kinetics of nucleation and crystallization of PCL revealing details about the ordering processes (nucleation, crystal formation, crystallization, cold crystallization, and crystal perfection) and the variations in glass transition of the remaining amorphous phase [22,23]. In another study, the structure development during crystallization of PCL was investigated by means of a comparison of viscoelastic, calorimetric, and optical observations, demonstrating that the viscoelastic properties start to improve at a relatively high degree of crystallinity [24].

In recent years, we assisted in the extensive interest in preparation of composite materials, consisting of different types of fillers embedded in a polymer matrix, in order to improve the properties of neat polymer. Consequently, the influence of filler on the crystallization behavior for PCL-based composites was thoroughly studied.

In these studies, the isothermal crystallization of nanocomposites based on PCL and clay at different filler amounts exhibited well-dispersed clay acting as a nucleating agent in PCL, in particular, decreasing the crystallization half-time. Moreover, the addition of the clay influenced both the bulk crystallization process and the spherulitic growth [25,26,27].

The crystallization of PCL composites containing carbon allotropes, such as multi-walled carbon nanotubes [28,29,30,31] and graphite nanoplatelets [32], and fillers from natural resources, such as bamboo cellulose [33], starch [34], nanocrystalline cellulose [35,36], and macaìba fibers [37], has been studied in detail.

Moreover, the addition of inorganic fillers, such as glass particles [38], barium sulphate [39], iron oxide nanoparticle [40], alumina, and niobium pentoxide [41], proved to be an efficient solution to enhance the crystallization rate of PCL in the composites.

The surface modification of inorganic fillers by means of chemical treatment with silanes led to a heterogeneous nucleation to PCL crystallization; in particular, surface treatment of basalt fibers [42] and halloysite nanotubes [43] strongly enhances the nucleation activity of the polymer matrix.

The present work reports on structural characteristics and thermal properties of PCL composites prepared with hollow glass microspheres (HGM) or silanized hollow glass microspheres (HGMf) suitable for RM technology. In particular, this paper describes the consequence of functionalization and amount of microspheres on the isothermal and non-isothermal crystallization kinetics of PCL.

## 2. Materials and Methods

### 2.1. Materials

Poly(ε-caprolactone) (PCL) CAPA 6800, with an average molecular weight of 84,000 g/mol, a density of 1.15 g/cm^3^, and a melt flow index of 3 g/10 min (160 °C, 2.16 kg) was selected as the matrix and purchased from Perstorp (Malmö, Sweden). The sodium borosilicate hollow glass microspheres (HGM) iM16K, with a density of 0.46 g/cm^3^, average diameter of 20 µm, and crush strength of 16,000 psi, were used as filler and supplied by 3M (Saint Paul, MN, USA). The coupling agent used for filler modification was an APTES [(3-aminopropyl)triethoxysilane] (purity ≥ 98.5%) supplied by Sigma-Aldrich (Milan, Italy).

### 2.2. Hollow Glass Microspheres Surface Functionalization

The surface functionalization of HGM consists of two steps: hydroxylation and silanization reaction. In the first step, 10 g of HGM were added to an alkaline solution of water (400 mL) and NaOH (0.5 mol/L). The obtained suspension was stirred for 1 h at 90 °C. Subsequently, the solution was filtered off and the HGM were washed with water until the pH of the water achieved neutrality. Then, residual water present on HGM was removed by drying in an oven at 70 °C for 8 h under vacuum. The silanization reaction was conducted adding the hydroxylated HGM to a solution of 2 g APTES in 200 mL ethanol with 0.2 g n-propylamine as catalyst. The silanization was carried out under reflux at 70 °C for 1 h. The functionalized hollow glass microspheres (HGMf) were filtered off and washed with ethanol and water to eliminate unreacted/ungrafted silane. Lastly, HGMf were dried at 80 °C for 24 h under vacuum [44]. The effectiveness of the functionalization was proven by Fourier transform infrared spectroscopy in our previous work [19].

### 2.3. Composite Processing

Composites of PCL were manufactured in a laboratory internal mixer Brabender (Duisburg, Germany) Plasticorder AEV 153. All composites containing various amounts of pre-dried at 100 °C for 1 h of untreated or functionalized microspheres were processed at 100 °C for 10 min with a rotor speed of 60 rpm under a nitrogen atmosphere. The neat polymer was also processed in the internal mixer under the same conditions for reference purposes. Afterwards, the materials were compression molded in disc shape (diameter 25 mm, thickness 0.5 mm) by using a laboratory press at 100 °C under 60 bar for 3 min. Finally, the samples were water cooled to room temperature.

### 2.4. Methods

Scanning Electron Microscopy (SEM) of the composites was performed using a benchtop Hitachi (Tokyo, Japan) TM3000 instrument. SEM analysis was performed on cross-sections of the samples cryo-fractured in liquid nitrogen. The samples were sputter coated with a thin layer of gold for good conductivity of electron beam prior observation under the SEM.

Differential Scanning Calorimetry (DSC) analysis performed by a Perkin Elmer (Waltham, MA, USA) DSC 8000 under nitrogen atmosphere. Isothermal crystallization experiments were conducted according to the following procedure. The sample was kept at 110 °C for 5 min to erase the thermal history. Afterwards, the sample was cooled at 80 °C/min to the selected isothermal crystallization temperature, *T_c_*. The heat flow of the isothermal crystallization was acquired as a function of time. Non-isothermal crystallization experiments were performed as follows. The sample was kept for 5 min at 110 °C in order to allow the complete melting; after that, the sample was cooled to room temperature at different cooling rates: 7.5, 10, 15, and 20 °C/min. In the final scan, the sample was heated to 100 °C with a heating rate equal to 10 °C/min to evaluate the melting temperature *T_m_*. Three separately weighed samples from each composition were analyzed by DSC.

Wide Angle X-ray Diffraction (WAXD) data were collected at 18 °C using a Siemens (Munich, Germany) D-500 diffractometer provided with a Siemens FK 60-10 2000 W tube (Copper K_α_ radiation, *λ* = 0.154 nm), operated at 40 kV and 40 mA. The data were collected from 5 to 40 2*θ*° with a step size of 0.02 2*θ*° intervals.

Small Angle X-ray Scattering (SAXS) measurements were carried out at 18 °C with a Kratky Compact Camera. Monochromatized Cu Kα radiation (*λ* = 0.154 nm) was produced by a Siemens Krystalloflex 710 generator equipped with a Siemens FK 60-10 2200 W Cu target tube operated at 40 kV and 40 mA. The scattered intensity was counted by using a step scanning proportional counter and the abscissa variable was *h* = sin(*θ*) 4π/*λ*. The blank scattering was subtracted from the sample scattering after correction for absorption and desmeared. The pseudo-two-phase model was applied to analyze the experimental scattering data according to the Vonk approach, as described in a previous paper [45].

## 3. Results and Discussion

### 3.1. Processing and Morphology

Samples of PCL and PCL-based composites were prepared in an internal mixer, as described in the experimental section, and coded as reported in Table 1.

Fragile fracture surfaces of the composites were studied by SEM to investigate the distribution and dispersion of microspheres within the polymer matrix and their adhesion to it. The micrographs of representative composite samples are shown in Figure 1.

In general, microspheres are undamaged after the mixing process, highlighting that their high crush strength allowed the withstanding of the mechanical stresses of the mixing process. The PCL-based composites exhibit a uniform distribution and a good dispersion of the fillers; thus, aggregates and agglomerates of fillers are not present.

However, the samples filled with HGM show a weak adhesion of the microspheres to PCL, as highlighted by the spherical cavities caused by the HGM pull-out and the presence of free microspheres on the sample’s surface (Figure 1a). PCL-HGMf composites showed a better adhesion testifying that the silanization of fillers enhances the chemical affinity and interactions with polymer matrix (Figure 1b).

A reduction in the cavities due to the pull out of the microspheres is observed for PCL-HGMf composites. Moreover, the cavities exhibit the residual part of the microspheres adhered to their walls before the cryo-fracture. Particularly, the functionalized microspheres result in being more wetted by the polymer matrix and well adhered to it, and thus some of them cracking during cryo-fracture.

### 3.2. Isothermal Crystallization by DSC

PCL and PCL-based composites were crystallized at various *T_c_*, that is 42, 43, 44, 45, and 46 °C, and the effect of the *T_c_* on the evolution of relative crystallinity (*X_r_*) versus time (*t*) was evaluated (Figure 2 and Appendix A).

The half-time of crystallization (*t_1/2_*), defined as the time at which the extent of crystallization is completed to 50%, is reported in Figure 3. In the investigated temperature range, the crystallization kinetic of all the samples enhances with the decrease in *T_c_*.

The presence of HGM caused a decrease in *t_1/2_* and the rate of crystallization increased with enhancing the HGM content in the PCL-HGM composite series (Figure 3a). Moreover, the composites containing the functionalized microspheres showed a faster crystallization rate compared to PCL-HGM at the same filler amount (Figure 3b), as a result of the improved dispersion of the HGMf in the polymer matrix due to the enhanced affinity between the phases [42,43].

The theory of Avrami was used to analyze the evolution of the relative crystallinity, *X_r_*, with time:(1)$$X_{r}\left(t\right)=1-\exp(-{K\;t\;}^{n})$$
where *n*, i.e., the Avrami exponent and *K*, i.e., the growth function, are parameters depending on the type of nucleation and on the geometry of the growing crystals [46]. The Avrami constants *K* and *n* for were calculated from the intercept and the slope, respectively, of the straight lines obtained by plotting log[–ln(1 − *X_r_*)] versus log *t* (Figure 4 and Appendix A) and are reported in Table 2.

In the range of investigated crystallization temperatures, pure PCL exhibits a straight line with a good correlation for *X_r_* values between 0.05 and 0.95. A fractional value of the Avrami exponent ranging from 3.2 to 3.9 was obtained, indicating a three-dimensional crystal growth. Similar values of the *n* parameter were found in the literature as a function of the PCL molecular weight and crystallization temperature [47,48].

All the analyzed composites showed an initial linear behavior, similarly to neat PCL, followed by a change in the slope at the end of the crystallization process, suggesting the presence of a two-step crystallization process [25,48,49]. The change in the slope appears at approximately 0.75 of relative crystallinity, pointing out a reduction in the growth rate of crystallization. Therefore, we evaluated different values of Avrami constants from the degree of crystallinity onwards, named *n’* and *K’* (Table 2). The decrease in the Avrami parameter, (*n’* values about 2) indicates differences in the mechanism of nucleation and the formation of near crystallites that hinder the process of ongoing crystallization.

The parameters *K* and *K’* decrease by increasing the *T_c_*. Moreover, the growth function value increases in the presence of filler, testifying to the nucleation effect of the glass microspheres in the PCL composites. This effect depends on the filler content and is more remarkable in the presence of the silanized filler.

### 3.3. Non-Isothermal Crystallization by DSC

The non-isothermal crystallization of PCL from the melt in the presence of HGM and HGMf was deeply investigated. Figure 5 depicts the DSC scans for the crystallization of PCL at different cooling rates. As expected, the crystallization peak temperature (*T_p_*) shifted to lower temperatures with enhancing cooling rate.

The variation of *T_p_* as a function of the cooling rate for PCL and PCL composites is presented in Figure 6. In general, the composites showed higher *T_p_* values than pure PCL. Moreover, the PCL-HGMf composites crystallized at slightly higher temperatures than the PCL-HGM composites due to the increased filler-matrix interactions and improved dispersion of HGMf in the PCL matrix. Similar results were reported by Bikiaris et al. [43], where the loading of a silanized filler to the PCL matrix resulted in an increase in the *T_p_* of the polymer matrix. Moreover, the nucleating effect of glass microspheres has been observed on several polymer matrices as well as polybutylene succinate [50,51,52], polypropylene [53,54], and polylactide [55].

The nucleation activity of the untreated and silanized glass microspheres in the PCL matrix was analyzed by the theory of Dobreva and Gutzow [56,57]. This model quantifies the nucleation activity factor (*φ*) of a filler during the non-isothermal crystallization of the polymer from the melt. The nucleation activity factor is near to zero (*φ* ≈ 0) when the filler is strongly active, while it tends to be near one (*φ* ≈ 1) for the inert filler. At temperatures close to melting, the cooling rate for nucleation from the melt can be expressed by the following equation:(2)$$\ln\;\beta \;=\mathrm{const}\;-\;\frac{B}{\Delta T_{p}^{\;2}}$$
where *β* is the cooling rate, Δ*T_p_* = *T_m_* − *T_p_* is the undercooling degree and *B* is a parameter related to the three-dimensional nucleation. The plots of ln *β* versus 1/Δ*T_p_*^2^ are displayed in Figure 7 for neat PCL and all the composites under study. Straight lines were obtained for each sample and the values of *B* for the pure PCL and PCL composites were calculated from the slope.

Then, *φ* values shown in Figure 8, were calculated by:(3)$$\varphi =\frac{B_{\mathrm{comp}}^{\;}}{B_{\mathrm{PCL}}^{\;}}$$
where *B*_comp_ and *B*_PCL_ are the *B* values for PCL-based composite and pure PCL, respectively. As already mentioned, the lower the *φ* factor, the higher the nucleation activity. It is evident from Figure 8 that both the HGM and HGMf accelerate the PCL crystallization process, which results in a reduction in process cycle times during rotomolding. In particular, the PCL-HGMf composites exhibit higher nucleation activity in the case of the sample loaded with 20 wt% HGMf being more active (*φ* = 0.52), as a consequence of good dispersion and strong interactions, as observed by mechanical and morphological analysis [19].

### 3.4. Structural Characterization

The crystalline and supermolecular structure of pure PCL and composites loaded with 20% of HGM or HGMf, was analyzed by X-ray techniques. The structural data of samples crystallized from the melt in (i) isothermal conditions at 46 °C or, (ii) non-isothermal conditions by fast cooling at 80 °C/min, are showed in Table 3 and some selected diffractograms are reported in Figure 9.

The WAXD profile of the isothermal crystallized samples exhibits the characteristic peaks of the crystallographic planes (110), (111), and (200) of the poly(**ε**-caprolactone) orthorhombic cell. The presence of diffraction relative to the (111) plane is not observed for the samples crystallized in non-isothermal conditions as a consequence of the high rate of crystallization. A lower index of crystallinity (*X_c_*), reported in Table 3, was calculated for the samples crystallized in non-isothermal conditions.

Figure 10 shows the Lorentz-corrected plots for the samples isothermally crystallized at 46 °C. The primary peaks of the composites have no evident shift in respect to the peak of the pure PCL, indicating that the presence of the HGM or the HGMf do not affect the long period (*L_p_*) of PCL. Analogously, the samples crystallized in non-isothermal conditions show a similar behavior with minor values of long period (Table 3).

The lamellar structure is evaluated accounting for the pseudo-two-phase model. The calculated structure parameters, that is the long period (*L*), the average lamellar thickness (*t_c_*), and the thickness of transition layer (*E*), are reported in Table 3. The *L* values confirm the trend observed for the *L_p_* periodicities, while in both adopted crystallization conditions, a lower value of the lamellar thickness was calculated for the composites compared to pure PCL.

## 4. Conclusions

PCL-based composites, already proven to be rotomoldable in previous work, were prepared by melt blending using untreated and silanized glass microspheres. Morphological analysis, performed by SEM, showed a uniform distribution and a good dispersion of the glass microspheres into the polymer matrix. In the presence of HGMf, a high wettability of the filler and strong adhesion between the filler and matrix was observed due to the enhanced chemical affinity that was in turn obtained by means of the silanization treatment.

An exhaustive study on the crystallization behavior of the prepared materials was carried out through thermal and structural analysis. In general, calorimetric study highlighted the nucleating effect of glass microspheres during both isothermal and non-isothermal crystallization, well testified by the half-time lowering of crystallization and the increase in the crystallization peak temperature, respectively. In particular, the nucleating effect of filler and the consequent faster crystallization was more markedly observed for the PCL-HGMf composites because of an optimal dispersion and stronger filler-matrix interactions. The theory of Avrami was applied to analyze the evolution of the relative crystallinity during isothermal crystallization. The obtained results showed a three-dimensional crystal growth during the entire crystallization process for neat PCL, while a two-step crystallization process was observed for PCL-based composites, which exhibited a first step similar to neat PCL followed by a second step characterized by a reduction in the growth rate of crystallization due to structural limitations during the crystallization. The nucleation activity of the glass microspheres within the polymer matrix during the non-isothermal crystallization was evaluated using the theory of Dobreva and Gutzow. The nucleation activity was found to be dependent on the filler content and resulted in being higher in the case of PCL-HGMf composites. Differences in the structural parameters for PCL-based materials were induced by the microspheres type, untreated or functionalized ones, and the adopted crystallization conditions.

In conclusion, this work provides useful information on the role of glass microspheres in the crystallization of PCL, drawing attention to their nucleating effect that results in being fundamental to polymer process technologies, such as the widespread rotational molding technology.

## Figures and Tables

**Figure 1 polymers-14-04326-f001:**
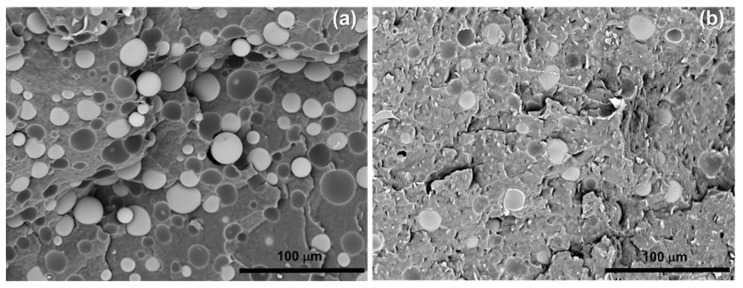
SEM image (magnification 600×) of PCL–HGM20 (**a**) and PCL–HGMf20 (**b**).

**Figure 2 polymers-14-04326-f002:**
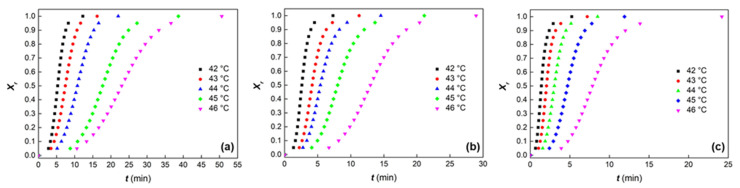
Relative crystallinity as a function of time for PCL (**a**), PCL-HGM10 (**b**), and PCL-HGMf10 (**c**) under isothermal conditions at different *T_c_*.

**Figure 3 polymers-14-04326-f003:**
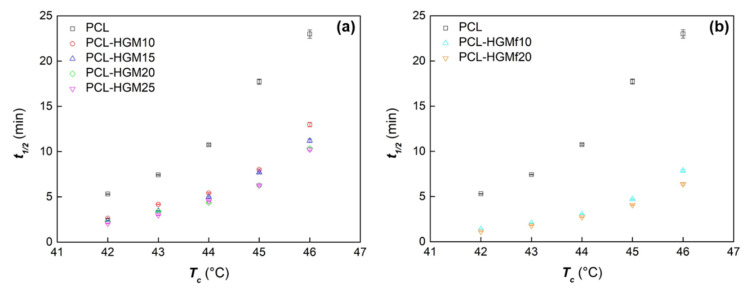
Half-time of crystallization vs. the crystallization temperature (average values with standard deviation) for PCL and composites PCL-HGM (**a**) and PCL-HGMf (**b**).

**Figure 4 polymers-14-04326-f004:**
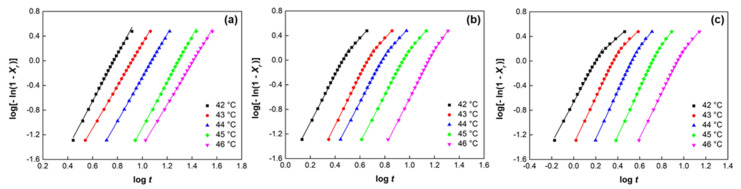
Plots of log[−ln(1 − *X_r_*)] vs. log *t* for isothermal crystallization of PCL (**a**), PCL-HGM10 (**b**), and PCL-HGMf10 (**c**) at different *T_c_*.

**Figure 5 polymers-14-04326-f005:**
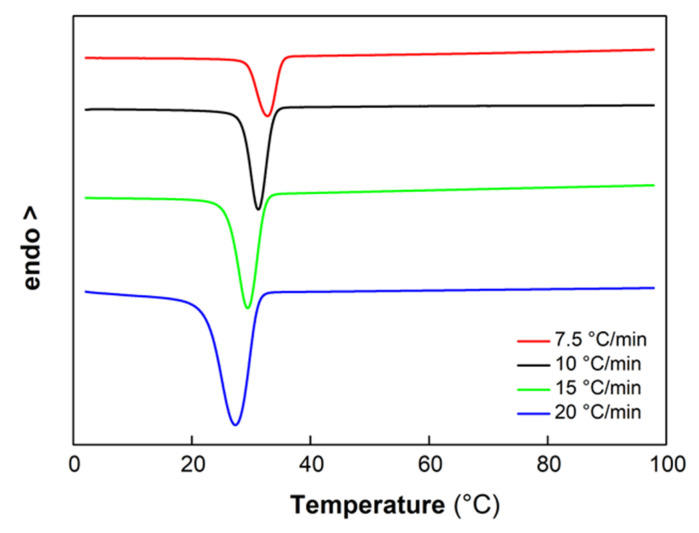
DSC cooling scans from the melt for PCL at different cooling rates.

**Figure 6 polymers-14-04326-f006:**
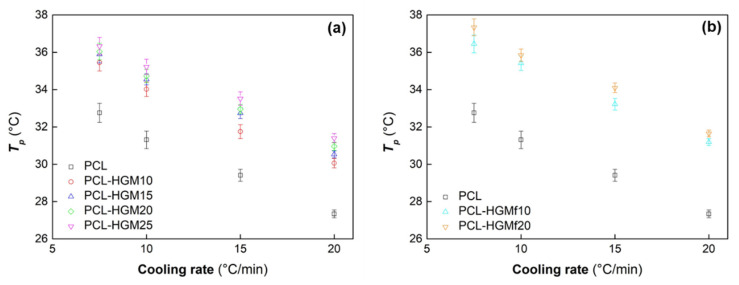
Peak temperatures (average values with standard deviation) for non-isothermal crystallization of PCL and composites PCL-HGM (**a**) and PCL-HGMf (**b**).

**Figure 7 polymers-14-04326-f007:**
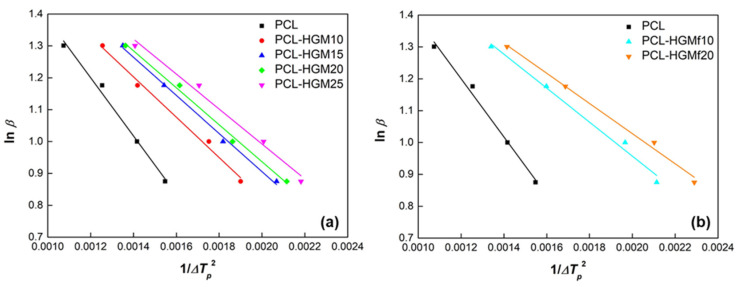
Dobreva plots for PCL and composites PCL-HGM (**a**) and PCL-HGMf (**b**).

**Figure 8 polymers-14-04326-f008:**
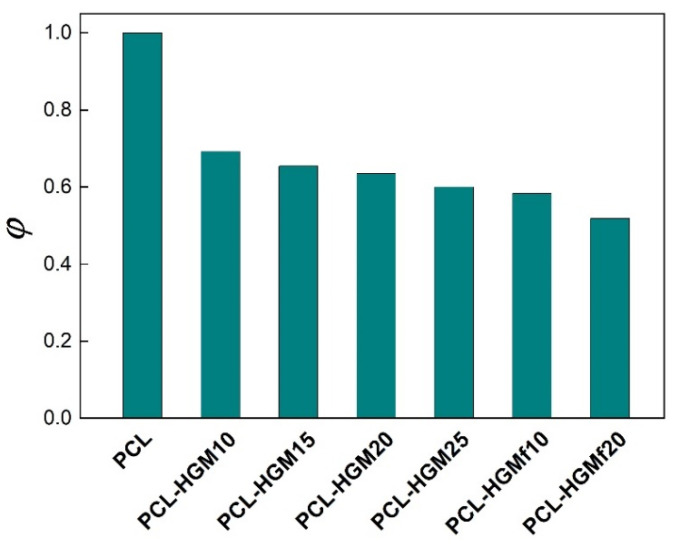
Nucleation activity factor for PCL and composites.

**Figure 9 polymers-14-04326-f009:**
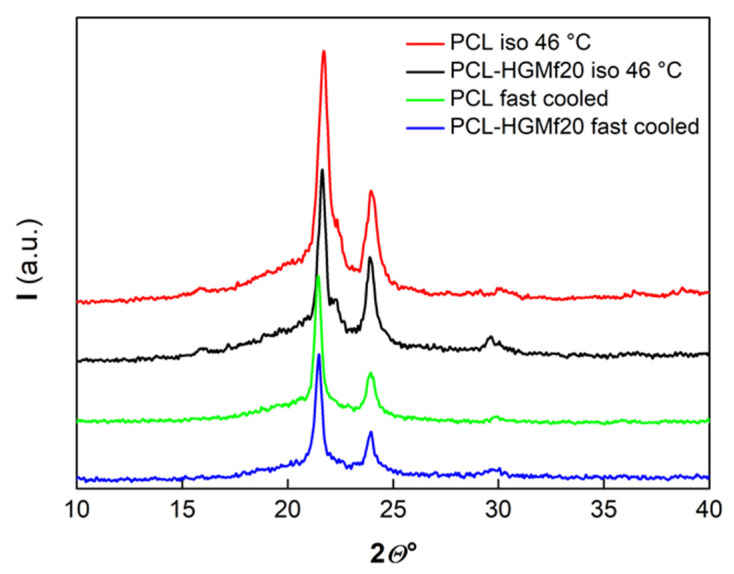
WAXD patterns for pure PCL and PCL–HGMf20 composite.

**Figure 10 polymers-14-04326-f010:**
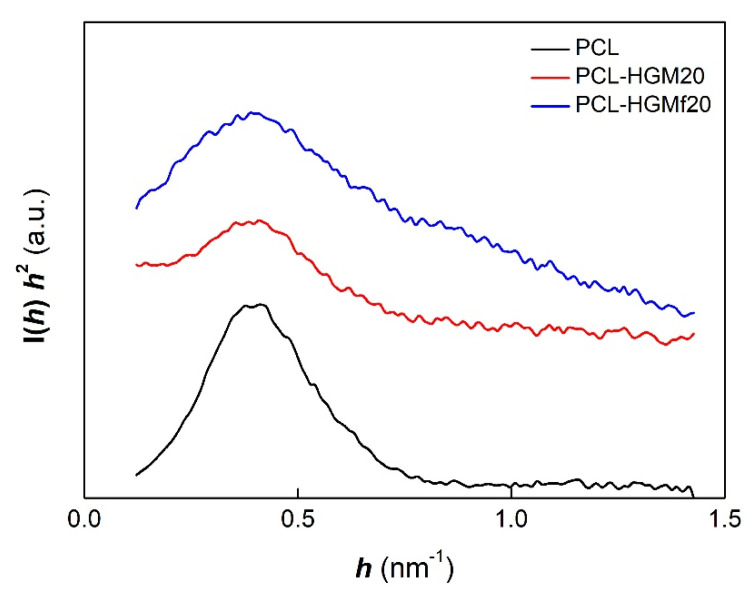
SAXS profiles for samples isothermally crystallized at 46 °C.

**Table 1 polymers-14-04326-t001:** Composition of the PCL-based composites.

Sample	PCL/Glass Microspheres (wt/wt)
PCL	100/0
PCL–HGM10	90/10
PCL–HGM15	85/15
PCL–HGM20	80/20
PCL–HGM25	75/25
PCL–HGMf10	90/10
PCL–HGMf20	80/20

**Table 2 polymers-14-04326-t002:** Half-time of crystallization and Avrami parameters of PCL and composites.

Sample		*T_c_* (°C)				
		42	43	44	45	46
PCL	*n*	3.8	3.4	3.5	3.7	3.4
	*K* (min^−1^)	1.2 × 10^−3^	7.0 × 10^−4^	1.6 × 10^−4^	1.6 × 10^−5^	1.8 × 10^−5^
PCL–HGM10	*n*	3.9	3.9	3.8	3.8	3.9
	*K* (min^−1^)	1.6 × 10^−2^	2.2 × 10^−3^	1.1 × 10^−3^	2.3 × 10^−4^	2.5 × 10^−5^
	*n’*	2.3	2.2	2.3	2.3	2.5
	*K’* (min^−1^)	9.6 × 10^−2^	3.7 × 10^−2^	1.9 × 10^−2^	6.5 × 10^−3^	5.3 × 10^−4^
PCL–HGM15	*n*	3.7	3.9	3.8	3.9	3.9
	*K* (min^−1^)	2.6 × 10^−2^	5.1 × 10^−3^	1.5 × 10^−3^	1.9 × 10^−4^	5.8 × 10^−5^
	*n’*	2.3	2.5	2.0	2.5	2.5
	*K’* (min^−1^)	1.2 × 10^−1^	4.1 × 10^−2^	3.6 × 10^−2^	5.0 × 10^−3^	2.3 × 10^−3^
PCL–HGM20	*n*	3.3	3.6	3.8	3.8	3.9
	*K* (min^−1^)	4.3 × 10^−2^	8.0 × 10^−3^	2.3 × 10^−3^	6.0 × 10^−4^	7.2 × 10^−5^
	*n’*	1.8	2.1	2.4	2.4	2.4
	*K’* (min^−1^)	1.9 × 10^−1^	7.0× 10^−1^	2.6 × 10^−2^	9.7 × 10^−3^	3.0 × 10^−3^
PCL–HGM25	*n*	3.7	3.7	3.9	3.8	3.7
	*K* (min^−1^)	5.8 × 10^−2^	1.2 × 10^−2^	1.8 × 10^−3^	6.3 × 10^−4^	1.0 × 10^−4^
	*n’*	1.7	2.2	2.1	2.6	2.2
	*K’* (min^−1^)	2.6 × 10^−1^	7.8 × 10^−2^	3.6 × 10^−2^	6.9 × 10^−3^	4.5 × 10^−3^
PCL–HGMf10	*n*	3.4	3.7	3.9	3.8	3.7
	*K* (min^−1^)	2.1 × 10^−1^	4.5 × 10^−2^	9.2 × 10^−3^	1.7 × 10^−4^	3.5 × 10^−4^
	*n’*	1.6	1.8	2.4	2.5	2.2
	*K’* (min^−1^)	5.7 × 10^−1^	2.6× 10^−1^	6.2 × 10^−2^	1.7 × 10^−2^	1.0 × 10^−2^
PCL–HGMf20	*n*	3.2	3.5	3.5	3.6	3.6
	*K* (min^−1^)	4.7 × 10^−1^	8.5 × 10^−2^	1.9 × 10^−2^	4.4 × 10^−3^	8.7 × 10^−4^
	*n’*	1.5	2.1	2.0	2.2	1.9
	*K’* (min^−1^)	8.1 × 10^−1^	2.4× 10^−1^	1.1× 10^−1^	4.2 × 10^−2^	2.7 × 10^−2^

**Table 3 polymers-14-04326-t003:** Structural and supermolecular parameters of pure PCL and composites.

Sample	*X_c_*	*L_p_* (nm)	*L* (nm)	*t_c_* (nm)	*E* (nm)
PCL iso 46 °C	0.36	15.8	14.6	5.5	0.3
PCL–HGM20 iso 46 °C	0.27	16.0	15.8	4.5	0.5
PCL–HGMf20 iso 46 °C	0.28	15.8	15.0	3.8	0.4
PCL fast cooled	0.32	14.0	13.0	4.4	0.3
PCL–HGM20 fast cooled	0.25	14.3	14.7	3.6	0.4
PCL–HGMf20 fast cooled	0.26	14.7	14.8	2.8	0.4

## Data Availability

Data is contained within the article or Appendix A.

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
