# Peer review of "Crystallization Behavior of Poly(ε-Caprolactone)-Hollow Glass Microspheres Composites for Rotational Molding Technology"

_polymers, 2022, doi:10.3390/polym14204326_

Round 1
Reviewer 1 Report
The manuscript can be published in Polymers after considering the following points:
- In Section 2.2, please demonstrate the microsphere functionalization process (hydroxylation and salinization) in a schematic, and clearly show the grafting sites and functional groups of the molecules.
- It is recommended to characterize the microspheres after hydroxylation and salinization to make sure that the microspheres are effectively functionalized. FTIR is suggested to do so.
- From Figure 1, it seems that the sizes of microspheres are different. If that is correct, please report the size distribution for microspheres used in composites.
- In the discussion section, please discuss how the obtained results and crystallization properties will benefit the rotational molding process and product (with regard to parameters such as composite brittleness, optical properties, etc.)
Author Response
Point 1: In Section 2.2, please demonstrate the microsphere functionalization process (hydroxylation and salinization) in a schematic, and clearly show the grafting sites and functional groups of the molecules.
Response 1: The reaction path for the functionalization of hollow glass microspheres particle surface was reported in our previous paper (Scheme 1, Vignali et al. Polymers 2019, 11, 624). We added the reference [19] in Section 2.2.
Point 2: It is recommended to characterize the microspheres after hydroxylation and salinization to make sure that the microspheres are effectively functionalized. FTIR is suggested to do so.
Response 2: The effectiveness of the treatment was proved by FTIR spectroscopy as presented in our previous work (Figure 2, Vignali et al. Polymers 2019, 11, 624), where we reported that the FTIR curves of both HGM show a characteristic peak between 1030 cm-1 and 970 cm-1 corresponding to the stretching of Si–O–Si bond. The FTIR spectrum of HGMf shows a peak at 1630 cm-1 corresponding to the –NH2 scissoring of primary amine groups. This characteristic peak confirms the anchoring of amine groups borne by APTES moieties onto the previously hydroxylated HGM surface. A sentence on the effectiveness functionalization was added to the revised manuscript in paragraph 2.2.
Point 3: From Figure 1, it seems that the sizes of microspheres are different. If that is correct, please report the size distribution for microspheres used in composites.
Response 3: The technical data sheet of the microspheres reports exclusively that the average diameter is 20 µm without the size distribution curve of HGM. However, SEM investigations showed the presence of microspheres with diameter between 10 and 30 µm.
Point 4: In the discussion section, please discuss how the obtained results and crystallization properties will benefit the rotational molding process and product (with regard to parameters such as composite brittleness, optical properties, etc.).
Response 4: The main advantage of obtained crystallization properties on rotational molding process is the reduction of process cycle times and we highlighted this feature in paragraph 3.3. As regards the mechanical properties of the products, we reported in our previous paper (Vignali et al. Polymers 2019, 11, 624) that the PCL–HGMf composites exhibit an increase in both the Young’s modulus and tensile strength compared to the neat PCL and composites containing unmodified microspheres.

Reviewer 2 Report
The paper is well written and I suggest to publish as it is.
Author Response
We thank the Reviewer for the positive comment.
Reviewer 3 Report
To improve the manuscript, the following changes are recommended:
1) In the Abstract section should be provided a few quantitative data on the main properties.
2) On page 1, lines 42-43, in the phrase “An important peculiarity of RM is that the heating and cooling rates utilized can cover a wide range” should be given the values for the range of the heating and cooling rates.
3) The size and shape of the PCL and PCL-based composites prepared in an internal mixer should be specified after cooling to the room temperature of the melt samples.
4) Referring to Table 1. Composition by weight of the investigated materials should be clarified if the designed compositions were presented in Table 1 and if the compositions of the realized composites were checked by experimental works. Some comments should be provided on the reason why only two functionalized composites (PCL–HGMf10 and PCL–HGMf20) were selected in this study, and PCL–HGMf15 and PCL–HGMf25 composites were not considered to compare the results for similar compositions of functionalized and nonfunctionalized composites (PCL–HGM10, PCL–HGM15, PCL–HGMf20, and PCL–HGM25).
5) In the caption of Figure 1. SEM image of PCL–HGM20 (a) and PCL–HGMf20 (b) should be specified the magnification of the SEM images. In Figure 1 should be shown all the developed composites to support the comment “The PCL-based composites exhibit a uniform distribution and a good dispersion of the fillers, thus aggregates and agglomerates of fillers are not present.”.
6) Referring to the phrase “In particular, the PCL-HGMf composites exhibit higher nucleation activity with the sample filled with 20 wt% HGMf being more active (φ = 0.52), as a result of great dispersion and stronger interactions” the comment on “great dispersion and stronger interactions” even should be reformulated or should be proved experimentally since it is expressed too generally, and the dispersion of the HGM and HGMf was performed in a similar way, but their weight content was different in the composites, as well as the filler-matrix interactions were not presented with quantitative data. The tendency of the nucleation activity factor for the PCL and PCL-based composites is clearly seen in Figure 8 but should be interesting to be also studied PCL–HGMf15 and PCL–HGMf25 composites, especially PCL–HGMf25 one, which could have a better nucleation activity than PCL–HGMf20 composite.
7) The scale values should be shown on the Oy axis for I (a.u.) from Figure 9 and I(h) h2 (a.u.) from Figure 10.
Author Response
Point 1: In the Abstract section should be provided a few quantitative data on the main properties.
Response 1: As suggested by Referee, we provided some quantitative data in the Abstract.
Point 2: On page 1, lines 42-43, in the phrase “An important peculiarity of RM is that the heating and cooling rates utilized can cover a wide range” should be given the values for the range of the heating and cooling rates.
Response 2: As suggested by Reviewer, we modified the sentence adding the range of the cooling rates (5-30 °C/min).
Point 3: The size and shape of the PCL and PCL-based composites prepared in an internal mixer should be specified after cooling to the room temperature of the melt samples.
Response 3: The prepared materials were compression molded into discs before the morphological, structural and thermal characterization. We added the details of preparation in the paragraph 2.3.
Point 4: Referring to Table 1. Composition by weight of the investigated materials should be clarified if the designed compositions were presented in Table 1 and if the compositions of the realized composites were checked by experimental works. Some comments should be provided on the reason why only two functionalized composites (PCL–HGMf10 and PCL–HGMf20) were selected in this study, and PCL–HGMf15 and PCL–HGMf25 composites were not considered to compare the results for similar compositions of functionalized and nonfunctionalized composites (PCL–HGM10, PCL–HGM15, PCL–HGMf20, and PCL–HGM25).
Response 4:
The check of the composition of the realized composites was carried out by TGA as presented in our previous work (Table 3, Vignali et al. Polymers 2019, 11, 624). In particular, the residue values reported in Table 3 confirmed that, for each sample, the actual content of filler was consistent with the nominal amount employed.
We considered to reinforce PCL with an opportune quantity of HGM in order to obtain lightweight composite materials: the addition of 20 wt % HGM implies a decrease of density by about 12% compared to neat PCL. So, we designed and prepared PCL-based composites with content of microspheres between 10 and 25 wt% that proved to be suitable for rotomolding in our previous paper (Vignali et al. Polymers 2019, 11, 624). Moreover, several works (e.g., Doumbia et al. Polym. Degrad. Stab. 2015, 114, 146–153.; Patankaret al. Compos. Part A Appl. Sci. Manuf. 2009, 40, 897–903; Patankar et al. Mater. Sci. Eng. A 2010, 527, 1361–1366) reported polymer composites filled with HGM in concentration between 10 and 30 wt%.
To evaluate the effect of the functionalized microspheres, we have decided to prepare only two PCL-HGMf composites with 10 and 20 wt% silanized HGM. Such contents of HGMf are enough to obtain a marked nucleating effect of the glass microspheres.
Point 5: In the caption of Figure 1. SEM image of PCL–HGM20 (a) and PCL–HGMf20 (b) should be specified the magnification of the SEM images. In Figure 1 should be shown all the developed composites to support the comment “The PCL-based composites exhibit a uniform distribution and a good dispersion of the fillers, thus aggregates and agglomerates of fillers are not present.”.
Response 5: As suggested by Reviewer, we specified the magnification (600x) of the SEM images in the caption of Figure 1. All the composites were observed by SEM to support the comment “The PCL-based composites exhibit a uniform distribution and a good dispersion of the fillers, thus aggregates and agglomerates of fillers are not present.” Micrographs of PCL-composites with 10wt.% of neat and silanized microspheres were reported in our previous work (Figure 5, Vignali et al. Polymers 2019, 11, 624).
Point 6: Referring to the phrase “In particular, the PCL-HGMf composites exhibit higher nucleation activity with the sample filled with 20 wt% HGMf being more active (φ = 0.52), as a result of great dispersion and stronger interactions” the comment on “great dispersion and stronger interactions” even should be reformulated or should be proved experimentally since it is expressed too generally, and the dispersion of the HGM and HGMf was performed in a similar way, but their weight content was different in the composites, as well as the filler-matrix interactions were not presented with quantitative data. The tendency of the nucleation activity factor for the PCL and PCL-based composites is clearly seen in Figure 8 but should be interesting to be also studied PCL–HGMf15 and PCL–HGMf25 composites, especially PCL–HGMf25 one, which could have a better nucleation activity than PCL–HGMf20 composite.
Response 6: The filler-matrix interactions were evaluated by morphological investigations, that showed as the HGMf were strongly wetted unlike HGM. In fact, the micrographs of PCL-HGM20 composites showed a gap between the microspheres and polymer matrix highlighting a weak interaction. As suggested by Reviewer, we reformulated the comment on “great dispersion and stronger interactions” in the paragraph 3.3.
The functionalization of glass microspheres with organic compounds enhances their chemical affinity with polymeric matrices, resulting in stronger interactions, homogenous dispersion and improved mechanical properties. As regards the mechanical properties of the products, we reported in our previous paper (Vignali et al. Polymers 2019, 11, 624) that the PCL–HGMf composites exhibit an increase in both the Young’s modulus and tensile strength compared to the neat PCL and composites containing unmodified microspheres. The morphological investigations, carried out by SEM (Figure 1) show that the HGMf are strongly wetted unlike HGM. In fact, the micrographs of PCL-HGM20 composites show a gap between the microspheres and polymer matrix highlighting a weak interaction. As suggested by Reviewer, we reformulated the comment on “great dispersion and stronger interactions” in the paragraph 3.3.
We did not prepared PCL–HGMf15 and PCL–HGMf25 composites; nevertheless, as suggested by Reviewer, PCL–HGMf25 could have a better nucleation activity than PCL–HGMf20 composite.
Point 7: The scale values should be shown on the Oy axis for I (a.u.) from Figure 9 and I(h) h2 (a.u.) from Figure 10.
Response 7: In Figure 9 and 10 the experimental curves are shifted for a better view, thus the scale values of the y axis are not reported (as usually presented in literature). The y axis are shown in arbitrary units (a.u.).
Round 2
Reviewer 1 Report
I overviewed the authors' manuscript previously published in Polymers (2019 Apr; 11(4): 624). I think the originality of the manuscript is under question. Also, the similarity index is 45% which is very high for a research paper. I think the authors need to clarify what makes this work original research significantly different from their previous work. Otherwise, I cannot recommend the manuscript for publication in Polymers.
Author Response
Reviewer 1:
"I overviewed the authors' manuscript previously published in Polymers (2019 Apr; 11(4): 624). I think the originality of the manuscript is under question. Also, the similarity index is 45% which is very high for a research paper. I think the authors need to clarify what makes this work original research significantly different from their previous work. Otherwise, I cannot recommend the manuscript for publication in Polymers."
Response:
The aims of our previous work (Polymers 2019, 11, 624) were: i) to prepare innovative PCL-based composites with different content of HGM, both untreated and silanized, ii) to test the processability (at laboratory and industrial scale) of the PCL–HGM composites via rotational molding technology; iii) investigate the effect of HGM on the thermal, rheological and mechanical properties. Moreover, the work reported that: “A comprehensive study on the nucleating effect of HGM on the PCL crystallization will be presented in a following work.”
The present work aims to deepen the study of PCL-HGM composites under the aspect of crystallization behavior, as the detailed knowledge of the crystallization process is fundamental to design PCL-based materials with tailored crystalline structure and properties. To the best of our knowledge, PCL-HGM composites have never been investigated through isothermal and non-isothermal crystallization studies. The work provides useful information on the role of glass microspheres (both treated and silanized) in the crystallization of PCL pointing the attention on their nucleating effect, which results fundamental in polymer process technologies, as the widely spread rotational molding. Moreover, we report about structural characteristics (crystalline and supermolecular structure) of PCL composites by using WAXD and SAXS techniques, highlighting differences in terms of crystallinity index and structural parameters as a function of the adopted crystallization conditions. All data reported in the present work are unpublished.
Therefore, we believe that this work is innovative and of great interest to the readers of Polymers.
Reviewer 3 Report
The manuscript is recommended for publishing since the authors performed a satisfactory revision.
Author Response
Reviewer 3:
The manuscript is recommended for publishing since the authors performed a satisfactory revision.
Response:
We thank the Reviewer for the positive final decision and useful revision.
Round 3
Reviewer 1 Report
-
Author Response
We thank the Reviewer for the suggestions and comments.